# Electrical Storm Induced by Cardiac Resynchronization: Efficacy of the Multipoint Pacing Stimulation

**DOI:** 10.3390/diseases12050105

**Published:** 2024-05-15

**Authors:** Anna Gonella, Carmelo Casile, Endrj Menardi, Mauro Feola

**Affiliations:** Department of Cardiology, Ospedale Regina Montis Regali, Strada del Rocchetto 99, 12084 Mondovì, Italy; anna.gonella@aslcn1.it (A.G.); carmelo.casile@aslcn1.it (C.C.); endrj.menardi@aslcn1.it (E.M.)

**Keywords:** resynchronization therapy, arrhythmia, multipoint stimulation

## Abstract

Although cardiac resynchronization therapy (CRT) reduces morbidity and mortality and reverses left ventricular (LV) remodeling in heart failure patients with LV electrical dyssynchrony, induced proarrhythmia has been reported. The mechanism of CRT-induced proarrhythmia remains under debate. In this case report, a description of how LV pacing induced polymorphic ventricular tachycardia immediately after the initiation of CRT has been reported. By changing the pacing configuration using a multipoint pacing stimulation, we can assume that induced ventricular tachycardia is related to the reentry mechanism facilitated by the unidirectional block. As a result, a multipoint pacing (MPP) configuration near the scar area can avoid the onset of a unidirectional block with the establishment of the reentry phenomenon, thus avoiding induced VTs.

## 1. Introduction

Cardiac resynchronization therapy (CRT) reduces morbidity and mortality and reverses left ventricular (LV) remodeling in heart failure patients with LV electrical dyssynchrony and systolic impairment [1,2]. CRT alone was demonstrated to decrease sudden death due to the remodeling properties on left ventricle and to reduce ventricular arrhythmias [3]. 

Despite this, CRT-induced proarrhythmia has been reported as a clinically serious, rare, and unpredictable phenomenon [4,5]. The mechanism of CRT-induced proarrhythmia remains under debate. The main reliable hypothesis seems to be that prolongation of the transmural dispersion of repolarization, by reversal of the activation sequence caused by LV epicardial pacing, might promote polymorphic ventricular tachycardia (VT) [5,6], while other theories speculate that re-entry is the possible responsible mechanism of induced proarrhythmia [7]. Furthermore, the electrical storm induced by CRT-pacing might be observed early after the implantation (3 days) [8], but a longer latent period (2 years) has also been described [9]. Independent of the delay of presentation, CRT-induced proarrhythmia is an uncommon, but life-threatening event after CRT implantation, causing electrical storm and worsening of hemodynamic conditions. The early presentation of proarrhythmia seemed to be clearly related to CRT pacing, while a delayed electrical storm might be easily attributed to the progression of cardiomyopathy and worsening of heart failure condition. However, in CRT-pacing patients in which a recovery of left ventricular function emerged, the BioCONTINUE study [10] clearly demonstrated that the risk of ventricular sustained arrhythmia proved to be significant (5.7%) at long-term (2-years) follow-up. In this case report, a description of how LV pacing induced polymorphic ventricular tachycardia immediately after the initiation of CRT has been reported.

## 2. Case Presentation

The patient is a 77-year-old man with systolic heart failure, depressed left ventricular function (30%) with an akinetic lateral wall, and functional class NYHA III. In his clinical history, chronic obstructive pulmonary disease emerged, as well as a persistent atrial fibrillation (AF), treated with cardioversion in 2019, and a multivessel coronary artery disease, treated with coronary angioplasty (PTCA) on the circumflex artery and the left anterior descending artery. In 2019, after recurrent ventricular tachycardia episodes, the patient was implanted with an implantable cardioverter defibrillator (ICD) for secondary prevention. He underwent placement of an Abbott Fortify Assura 1359QC device, with a Durata 7122Q right defibrillation lead. In 2022, during an hospitalization for symptoms of cardiac congestion related to an uncontrolled frequency in AF, an upgrade was performed, with the implantation of a cardiac resynchronization therapy defibrillator (CRTD), followed by an atrioventricular node (AVN) ablation in order to reduce the elevated frequency of AF, despite medical optimization (see Figure 1 for the correct position of electrode catheters after implantation). After AVN ablation, the device has programmed in VVIR to 85 bpm. During the night, caused by an initial delirium, quetiapine and haloperidol were administered in order to control the psychotic symptoms. The day after the CRTD implantation, polymorphic ventricular tachycardia was registered, which was initially treated intravenously with magnesium and amiodarone (see Figure 2). According to the hypothesis of proarrhythmic damage due to the new LV stimulation, a check of the device was performed, but the mechanism of CRT-induced proarrhythmia remained unknown during the LV stimulation (single site, D1-M1 configuration).

Although different poles were tested, the LV stimulation was turned off. After turning off the LV stimulation, the polymorphic ventricular tachycardia disappeared. In the absence of LV stimulation and during the antipsychotic therapy, the QTc interval was calculated at 424 ms. According to the clinical data available in the literature, which tested the hypothesis that pacing with a left ventricle lead positioned near the epicardial scar might be responsible of proarrhythmias, we decided to program the multipoint pacing stimulation (MPP) from the left side, thus modifying the generated wavefront and the refractory period (Figure 3). With the new wavefront generated with the multipoint stimulation, ventricular tachycardia/electrical storm no longer occurred. The patient discontinued taking a low dosage of amiodarone (600 mg/week), and bisoprolol was prescribed before hospital discharge (ECG at discharge, Figure 4). During a long-term follow-up (12 months), no other VT episodes were recorded in the memory of the device. 

Upper panel: activation map during BIV; lower panel: activation map during best MPP. BIV = biventricular pacing; MIPP = multipoint pacing. Printed with permission. 

## 3. Discussion

Potential pro-arrhythmic effects of CRT have been described in several previous experiences [4,5,6,7,8,9], in which biventricular and LV pacing are responsible for polymorphic and monomorphic VT, although the mechanism of CRT-induced TV remains unclear. These effects have been observed as VT induced by CRT in patients with ischemic and non-ischemic cardiomyopathy. Several authors [6] believe that changing the activation sequence near the scar areas, resulting from LV stimulation, is responsible for the unidirectional block, and reentry and might induce proarrhythmia. Roque et al. [8] have demonstrated that VT can be induced by stimulating the epicardial scar and managed with catheter ablation, identifying the intramyocardial reentry as the main responsible mechanism. Almost certainly, LV pacing near the scar seems to be the primary cause of inducted VT. In our patient, the LV lead position related to the scar observed in the previous coronary artery disease episodes seemed to be very close to the first configuration tested in the CRT stimulation (D1-M2).

It should be postulate that the presence of haloperidol and quetiapine in therapy might be play a role in the determinism of the proarrhythmic effect. Previous reports [11,12,13] underlined that the risk of torsades de pointes seemed was demonstrated, especially for women and for intravenously administration, particularly in regards to haloperidol. Less dangerous was the proven influence of quetiapine on QT length, which seemed to determine a QT shortening that might also be considered an indicator of the proarrhythmic effect. In our patient, the QTc interval during antipsychotic therapy was normal, the antipsychotic treatment was withdrawn quickly after the evidence of polymorphic VT, and no drugs were infused intravenously

Since the patient had a fairly wide scar, which also affected the proximal part of the catheter (M3-P4), we can assume that the change of the pole was not sufficient to avoid the unidirectional block and the initiation of the reentrant circuit. Our possible solution, then, was to try an extended and wider wavefront stimulation created by multipoint left ventricular pacing (MPP). The MPP is obtained when multiple pacing stimuli are delivered via a single quadripolar lead placed in a branch of the coronary sinus to achieve biventricular pacing [14]. Recent clinical single-center trials [14,15] suggest that MPP stimulation seems to acutely improve the measures of dyssynchrony, the hemodynamic response obtained through a recruitment of a larger portion of the left ventricle, resulting in a more homogenous and flat propagation (see Figure 3, related to Ref. [15]). Asvestas et al. [9] described that MPP stimulation was able to suppress a proarrhythmic effect in a CRTD patients 2 years after the implantation of the device, programming a larger front of stimulation (bipolar pacing: proximal 4 to RV coil). In the experience of Roque et al. [8], eight patients exhibiting the proarrhythmic effect of CTRD, in which a positive LV lead/scar relationship (i.e., defined as a lead tip positioned on a scar/border zone) were successfully treated by catheter ablation of the ventricular tachycardia. Furthermore, the possibility of cardiac resynchronization using His-bundle pacing, alone or in association with coronary sinus pacing, seemed to achieve a comparable prognostic effect in comparison to conventional CRT in HF patients [16,17]. 

In our patients, we decided to reactivate LV pacing using two different poles of the LV lead, activated simultaneously (D1-M2–M3-P4). After this change, the patient was monitored, and no VTs were recorded or treated (Figure 4). In our point of view, the wider wave front generated by the MPP near the scar area activated a larger part of the left ventricle, determining a synchronization of the tissue. We can assume, in our experience, that induced VTs are related to the reentry mechanism, facilitated by the unidirectional block. As a result, the MPP pacing configuration near the scar area can prevent the onset of a unidirectional block with the establishment of the reentry phenomenon, thus avoiding induced VTs.

Finally, our clinical experience underlined that left pacing may determine a ventricular remodeling, and this can certainly lead to a clinical benefit for the patient. However, the detection area of the lead must also be carefully evaluated in relation to a possible scar. In our case, MPP stimulation definitely proved to be an excellent solution for a beneficial patient outcome. 

## 4. Conclusions

In patients implanted with CRT, the possibility of an arrhythmogenic effect of cardiac stimulation should be identified. Patients at risk could be identified using imaging techniques (such as cardiac MRI, which can uncover areas of transmural myocardial fibrosis). Furthermore, the influence of LV lead position in determining the proarrhythmic effect was clearly documented in the MADIT-CRT study [18], in which an anterior LV lead position was associated with a higher risk of arrhythmic events compared to a posterior or lateral position. The explanation seemed to derive from the increased presence of myocardial infarction in patients in whom the stimulation of the left ventricle was localized in the anterior wall. Finally, CRT patients who experience an increase in ventricular arrhythmias soon after device implantation should be hospitalized for a longer period to monitor these events.

## Figures and Tables

**Figure 1 diseases-12-00105-f001:**
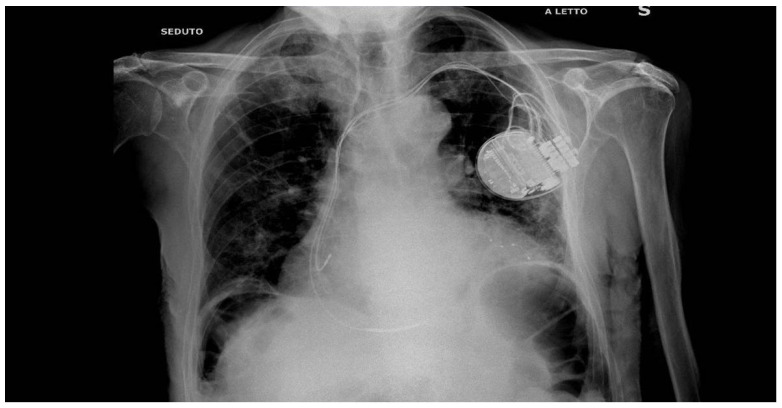
Thoracic Rx scan after the CRTD implantation: see the LV electrode catheter in a lateral vein of the coronary sinus. A coronary angioplasty of the MO branch of the left circumflex artery has been described in the history of this patient.

**Figure 2 diseases-12-00105-f002:**
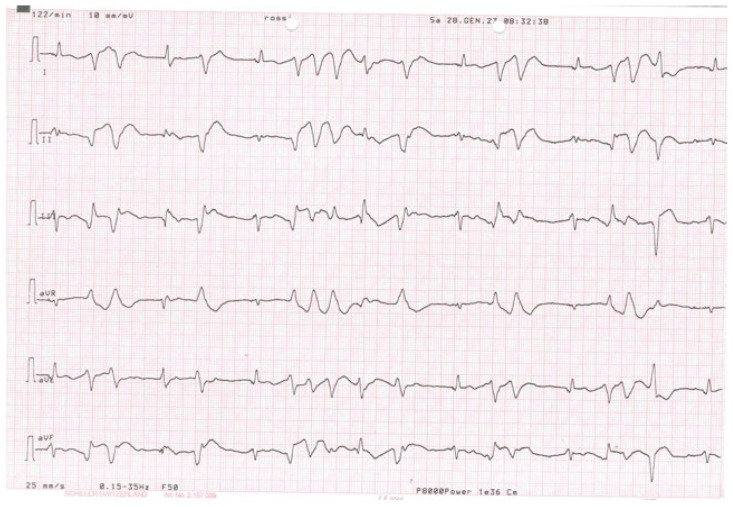
The polymorphic arrhythmic activity in the six-lead electrocardiogram 2 days after implantation.

**Figure 3 diseases-12-00105-f003:**
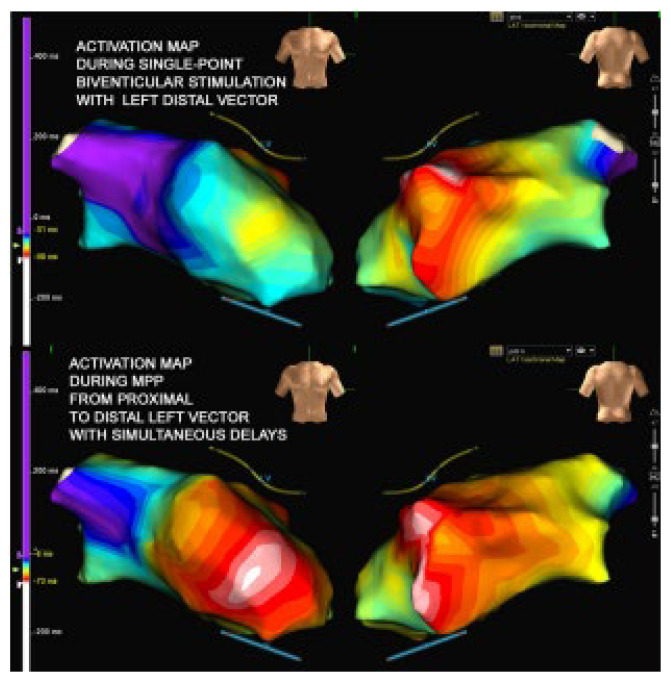
Different pulse propagation during biventricular pacing and MPP: MPP realizes the recruitment of a greater portion of the left ventricle.

**Figure 4 diseases-12-00105-f004:**
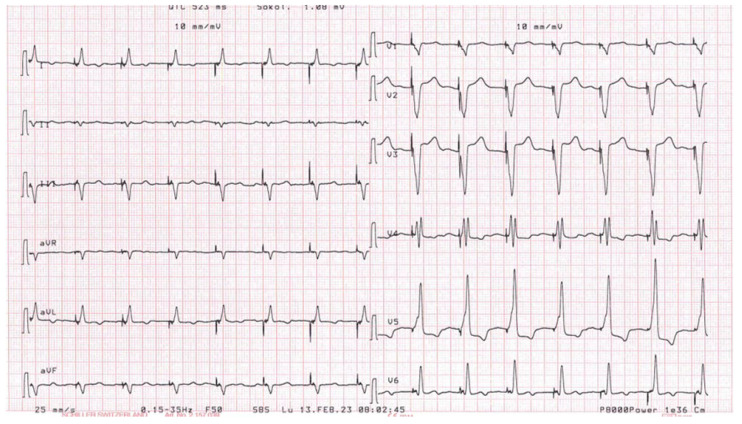
A 12-lead electrocardiogram after reprogramming the MPP of the device. An absence of ventricular arrhythmias emerged.

## Data Availability

Data are contained within the article.

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
