# Peer review of "Electrical Storm Induced by Cardiac Resynchronization: Efficacy of the Multipoint Pacing Stimulation"

_diseases, 2024, doi:10.3390/diseases12050105_

Round 1
Reviewer 1 Report (Previous Reviewer 3)
Comments and Suggestions for Authors
I still have major concerns regarding the tracings presented and their interpretation. Most of my considerations could not be addressed properly.
Other causes of T.d.P cannot be excluded as well.
Comments on the Quality of English Languageminor editing
Author Response
Thank you very much for taking the time to review this manuscript.
It's true, other causes cannot be certainly excluded but in line 110 we explained the reasons that led us to consider more probable the role of LV stimulation
Kind regards
Reviewer 2 Report (Previous Reviewer 4)
Comments and Suggestions for Authors
Manuscript definitely improved after author's corrections. Congratulations to the authors for theri very good paper, that could be accepted
Author Response
Thank you very much for taking the time to review this manuscript and for the compliments for our paper.
Kind regards
Reviewer 3 Report (New Reviewer)
Comments and Suggestions for Authors
The manuscript by Gonella et al., entitled "Electrical Storm Induced by CRT: Efficacy of Multipoint Pacing," describes a case report about a patient who underwent a CRT upgrade and was treated for ventricular arrhythmias the day after the upgrade. The authors suggest that an electrical storm took place due to pacing in scar tissue, which was resolved after reprogramming the CRT device to multipoint pacing.
While the case itself is highly relevant and interesting, the manuscript still requires some careful rewriting and editing, particularly regarding the definitions and the presentation of the figures. I would also suggest that the authors review the definitions of electrical storm and be more careful in their conclusions, as they do not (yet) provide evidence of the pacing electrode position in scar tissue. Additionally, the patient also received antipsychotics that can influence repolarization time and are potentially proarrhythmic. I believe that the ventricular arrhythmias in this patient may therefore have multifactorial causes.
Specific comments:
- Line 30: “polymorphic TV” should I think be “polymorphic VT”.
- Spell out the abbreviations when using it for the first time.
- Spell out and elaborate briefly (for the readers who are not familiar with this) about the BioCONTINUE study in line 40.
- End the introduction with the purpose of this manuscript/paper.
- Line 48: was the patient optimally revascularized?
- Line 56: spell out NAV ablation.
- Line 50: ventricular tachycardia is usually known as VT and not TV
- Line 49: where these VT episodes monomorphic or polymorphic? And what was the primary cause? Congestion?
- So the patient received quetiapine and haloperidol. What was the QT time prior and post this medication?
- The LV lead position appears to me more lateral or anterolateral on the chest X ray instead of inferolateral.
- Figure 2: please also show the 12-ECG prior upgrade to CRT.
- So the patient received a His Ablation due to persistent atrial fibrillation? What was his underlying rhythm? Was it a high nodal escape rhythm, because I see small QRS complexes in Figure 2. Was the patient pacing dependent?
- Line 75: the authors suggest that the LV lead was positioned near epicardial scar, please provide imaging, e.g. CMR or voltage mapping.
- Please specify figure 3 more with the views and anatomical positions of the (quadripolar) lead.
- The authors describe an electrical storm but on the ECG in figure 2 I can only appreciate short ventricular runs, and the patient was only treated with magnesium and betablocker but seem not to require sedation or cardioversion due to hemodynamic compromises. Can the authors more accurately describe how often the patient had these episodes within 24 hours? Please also consult the EHRA 2024 document on electrical storm for the accurate definitions.
Comments on the Quality of English LanguageRequires moderate linguistic editing as described in more detail to the authors.
Author Response
Thank you very much for taking the time to review this manuscript.
- Commnet 1: Line 30: “polymorphic TV” should I think be “polymorphic VT”.
Response 1: I don't understand this request
-Comment 2: Spell out the abbreviations when using it for the first time.
R 2: We have specified all the abbreviations present in the text.
- C. 3: Spell out and elaborate briefly (for the readers who are not familiar with this) about the BioCONTINUE study in line 40.
R 3: done
- C 4 End the introduction with the purpose of this manuscript/paper.
R4: we have described a case report
- C5: Line 48: was the patient optimally revascularized?
R5: yes see figure 1
- C6: Line 56: spell out NAV ablation.
R 6 done
- C7: Line 50: ventricular tachycardia is usually known as VT and not TV
R 7 done
- C8 Line 49: where these VT episodes monomorphic or polymorphic? And what was the primary cause? Congestion?
R8 (see line 75) After turning off the LV stimulation, the polymorphic ventricular tachycardia disappeared. In absence of LV stimulation and during the antipsychotic therapy the QTc interval was calculated 424ms. According to the clinical data available in literature which tested the hypothesis that pacing with a left ventricle lead positioned near the epicardial scar might be responsible of proarryhtmias
- C9: So the patient received quetiapine and haloperidol. What was the QT time prior and post this medication?
R9 see line 78
- C10 The LV lead position appears to be more lateral or anterolateral on the chest X ray instead of inferolateral.
R 10 the LV elettrocatheter is in a postero-lateral vein
- C11 Figure 2: please also show the 12-ECG prior upgrade to CRT.
R 11 unfortunately no other ECG are avaible
- C12: So the patient received a His Ablation due to persistent atrial fibrillation? What was his underlying rhythm? Was it a high nodal escape rhythm, because I see small QRS complexes in Figure 2. Was the patient pacing dependent?
R 12 see line 56
- C13 Line 75: the authors suggest that the LV lead was positioned near epicardial scar, please provide imaging, e.g. CMR or voltage mapping.
R 13: The scar could not be investigated with cardioMRI because the patient had already an ICD but the area of ​​akinesia at echocardiography was the lateral wall
- C14 Please specify figure 3 more with the views and anatomical positions of the (quadripolar) lead.
R 14 done
- C15 The authors describe an electrical storm but on the ECG in figure 2 I can only appreciate short ventricular runs, and the patient was only treated with magnesium and betablocker but seem not to require sedation or cardioversion due to hemodynamic compromises. Can the authors more accurately describe how often the patient had these episodes within 24 hours? Please also consult the EHRA 2024 document on electrical storm for the accurate definitions.
R 15: see line 62
Kind regards
Round 2
Reviewer 3 Report (New Reviewer)
Comments and Suggestions for Authors
The papers topic is relevant and made some improvements.
The reply to the reviewer is very limited.
The manuscript still contains a lot of spelling errors and the figures (especially the ECGs) are low resolution and not carefully aligned.
Other more content based comments:
- did the authors consider conduction system pacing to avoid the epicardial dispersion instead of multipoint pacing?
- the authors make bold statements in the conclusions about treatment recommendations, this cannot be stated based on a single case report.
Comments on the Quality of English LanguageThe manuscript needs extensive English language editing.
Author Response
Thank you very much for taking the time to review this manuscript.
We could not consider stimulation of the conduction system to avoid epicardial dispersion because the patient presented the arrhythmic problem arthermorefter CRT implantation. Furthermore, as suggested, we modified the conclusions by only proposing measures for patient management, not therapeutic solutions
Kind regard
This manuscript is a resubmission of an earlier submission. The following is a list of the peer review reports and author responses from that submission.
Round 1
Reviewer 1 Report
Comments and Suggestions for Authors
The grammar needs to be improved. As it is written, it is very difficult to understand the manuscript.
The case report makes statements about the scar tissue, but there is not evidence or imaging to support the location of the scar. It is not clear that it is near where the LV pacing leads are located, and therefore, the statement that MPP avoided the arrhythmia (of ventricular tachycardia) is not supported.
The patient also has other factors that could contribute to arrhythmia. Specifically, the anti-seizure medications he as taking. These medications were stopped at the same time the pacing was changed. Therefore, can we logically say that the end of the arrhythmia is due to the pacing change alone?
Comments on the Quality of English Language
The use of English is very difficult to understand. The grammar makes it difficult to understand what the authors mean.
Reviewer 2 Report
Comments and Suggestions for Authors
Thank you for the opportunity to review this case study. The case report is interesting but I have the following comments.
1. Grammar needs to be improved. For example Page 2; line 48: What does hypocinetic cardiomyopathy mean? Page 2; line 49: "Programmed" should be "performed"
English editing is recommended prior to publication.
2. How do the authors know that Multi-site pacing (i.e. change in pacing wavefront) was responsible for improvement in VT and not holding anti-psychotic drugs which prolong QT?
3. Quality of figure 3 needs to be improved
4. Introduction; lines 29-30: Consider adding following supportive reference concerning long term risk of ventricular arrhythmias (even with improvement in LVEF) after LV pacing/CRT which was shown BioContinue study
Gras, D., Clémenty, N., Ploux, S. et al. CRT-D replacement strategy: results of the BioCONTINUE study. J Interv Card Electrophysiol 66, 1201–1209 (2023).
Comments on the Quality of English Language
Consider editing by native language speaker
Reviewer 3 Report
Comments and Suggestions for Authors
I have concerns regarding the presented data, tracings, and their interpretation.
- T.d.P related to the added new drugs cannot be excluded. The authors should carefully assess QTc intervals before and after the procedure.
- Arrhythmias occurred following AVN ablation. There are many described cases of such arrhythmias following this ablation in patients used to high-rate AF. This is another confounding factor. How was the LRL programmed after ablation?
- In Figure 2: upper panel: there are short runs of PMVT but without clear pacing of QRS between! lower panel: the TdP episode seems to follow inappropriate pacing?! Authors should provide of EGM annotations at the episode onset and ICD sensing parameters.
- Figure 3 quality is not sufficient to be assessed!
- What is the origin of Figure 4 mapping? Did the author perform on the same patient?
- Figure 5: 12-lead ECG after MPP programming is not indicative of synchronized QRS. The QRS is pretty wide and there is BBS morphology in V1-V3, positive QRS in I aVL! Please clarify.
Comments on the Quality of English Languageshould be improved throughout the text.
Reviewer 4 Report
Comments and Suggestions for Authors
This paper is extremely interesting showing potential pro-arrhythmic effects of CRT in this case presentation. Congratulation to the authors for the very good case, I have only 2 minor comments in order to improve this case report: in discussion authors should simply cite the existence and the importance of physiologic cardiac pacing as CRT alternative and its potential impact on arrhythmic effects (DOI: 10.1111/pace.14336 ; and DOI: 10.1161/CIRCEP.123.012473). Please expand the topic and cite 2 suggested reference. Finally, a conclusion section for the paper is definitely mandatory